# Electrophysiological characteristics of non-pulmonary vein triggers excluding origins from the superior vena cava and left atrial posterior wall: Lessons from the self-reference mapping technique

**Yasuharu Matsunaga-Lee[1], Yasuyuki Egami[1], Sen Matsumoto[2], Nobutaka Masunaga[2], Kohei Ukita[1], Akito Kawamura[1], Hitoshi Nakamura[1], Yutaka Matsuhiro[1], Koji Yasumoto[1], Masaki Tsuda[1], Naotaka Okamoto[1], Masamichi Yano[1], Yuzuru Takano[3], Yasushi Sakata[4], Masami Nishino[1]*, Jun Tanouchi[1]**

1 Division of Cardiology, Osaka Rosai Hospital, Osaka, Japan, 2 Department of Cardiology, JCHO Hoshigaoka Medical Center, Osaka, Japan, 3 Department of Cardiology, Higashiosaka Citizen Hospital, Osaka, Japan, 4 Department of Cardiovascular Medicine, Osaka University Graduate School of Medicine, Suita, Japan

* 3522mn@gmail.com

## Abstract

### Background

The detailed electrophysiological characteristics of atrial fibrillation (AF) initiating non-pulmonary vein (PV) triggers excluding origins from the superior vena cava (SVC) and left atrial posterior wall (LAPW) (Non-PV-SVC-LAPW triggers) remain unclear. This study aimed to clarify the detailed electrophysiological characteristics of non-PV-SVC-LAPW triggers.

### Methods

Among 446 AF ablation procedures at 2 institutions, patients with reproducible AF initiating non-PV-SVC-LAPW triggers were retrospectively enrolled. The trigger origin was mapped using the self-reference mapping technique. The following electrophysiological parameters were evaluated: the voltage during sinus rhythm and at the onset of AF at the earliest activation site, coupling interval of the trigger between the prior sinus rhythm and AF trigger, and voltage change ratio defined as the trigger voltage at the onset of AF divided by the sinus voltage.

### Results

Detailed electrophysiological data were obtained at 28 triggers in 21 patients. The median trigger voltage at the onset of AF was 0.16mV and median trigger coupling interval 182msec. Normal sinus voltages (≧0.5mV) were observed at 16 triggers and low voltages (<0.5mV) at 12 triggers. The voltage change ratio was significantly lower for the normal sinus voltage than low sinus voltage (0.20 vs. 0.60, p = 0.002). The trigger coupling intervals were comparable between the normal sinus voltage and low sinus voltage (170ms vs. 185ms, p = 0.353).

**Data Availability Statement:** All relevant data are within the manuscript and its Supporting Information files.

**Funding:** The author(s) received no specific funding for this work.

**Competing interests:** The authors have declared that no competing interests exist.

## Conclusions

The trigger voltage at the onset of AF was low, regardless of whether the sinus voltage of the trigger was preserved or low.

## Introduction

Catheter ablation of atrial fibrillation (AF) has emerged as an effective treatment option and is currently an indication for symptomatic patients with drug refractory AF [1]. Pulmonary vein (PV) isolation is a cornerstone strategy to control AF by ablation [2] and additional substrate ablation failed to show superiority in non-paroxysmal AF patients [3]. Successful elimination of non-PV triggers is still important and contributes to better outcomes in paroxysmal [4] and non-paroxysmal patients [5]. Non-PV triggers from the superior vena cava (SVC) or left atrial posterior wall (LAPW) were usually treated with an isolation strategy. Non-PV triggers excluding origins from the SVC and LAPW (Non-PV-SVC-LAPW triggers) were considered to be related to worse clinical outcomes [6] because those triggers require precise mapping to eliminate them and are known to be difficult to eliminate [7].

Mapping of origins of non-PV triggers is usually performed using a 3-dimentional electro-anatomical mapping system. The techniques for the provocation or localization of non-PV triggers have been previously reported [8], but the detailed electrophysiological data has not been fully evaluated. We previously reported a new technique called self-reference mapping to map origins of non-PV triggers [9, 10]. This technique does not require any other fixed reference catheters and uses the previous earliest activation site as a reference. The operator repeatedly moves a PentaRay NAV (PEN) catheter (Biosense Webster Diamond Bar, CA, USA) to the earliest signal creating a new reference each time to map the non-PV triggers. This method has the advantage of a single-beat analysis, which creates a high-density map of non-PV triggers [11], and the detailed electrophysiological data could be analyzed. Therefore, the purpose of this study was to clarify the detailed electrophysiological characteristics of non-PV-SVC-LAPW triggers.

## Methods

### Study population

This study was a sub-analysis of the previously reported paper [10], in which the short- and long-term clinical outcomes of the self-reference mapping technique were shown. The study population has previously been reported [10]. In brief, we investigated 446 AF ablation procedures in 431 patients at Osaka Rosai Hospital and JCHO Hoshigaoka Medical Center from January 2017 to March 2019. Patients who had reproducible non-PV-SVC-LAPW triggers were enrolled. In this study, the detailed electrophysiological characteristics were analyzed while synchronizing the log reports of an electro-anatomic mapping system (CARTO 3, Biosense Webster Diamond Bar, CA, USA) and simultaneously recorded polygraph (LABSYSTEM™ PRO EP Recording System, Boston Scientific, Marlborough, MA, USA). All patients provided written informed consent for the AF ablation procedures and the use of their clinical data. The research protocol complied with the ethical guidelines of the Declaration of Helsinki in 1975. The institutional ethics committee of each site approved the study.

## AF ablation

All antiarrhythmic drugs were discontinued for at least 5 half-lives prior to the catheter ablation. Amiodarone was not used in this study group. Electrophysiological studies were performed under sedation with propofol and dexmedetomidine. Invasive arterial blood pressure monitoring, oxygen saturation measurements, and surface 12-lead ECGs were monitored during the entire procedure. The transseptal puncture was guided by intracardiac echocardiography to access to the left atrium (LA). All patients underwent a bilateral PV isolation using open irrigation contact catheters (ThermoCool Smart-Touch SF, Biosense Webster, Diamond Bar, CA, USA). The radiofrequency ablation power settings were 20–30 W for the posterior LA and 25–35 W for the anterior LA. After the PV isolation, the induction of non-PV triggers was performed. Additional ablation, including linear or an SVC ablation, was at the operator's discretion.

## Non-PV trigger induction and self-reference mapping of non-PV triggers

Our definition of non-PV triggers was AF initiating triggers. Frequent atrial premature complexes that did not induce AF were not defined as non-PV triggers. Non-PV triggers were induced with a continuous infusion of isoproterenol (ISP, 1–10 μg / min), rapid infusion of ISP (2–20 μg), and/ or adenosine triphosphate (ATP, 40-60mg). If AF was not initiated by the ATP and ISP infusions, sustained AF was induced by programmed atrial burst pacing. Non-PV triggers after restoration of sinus rhythm by intracardiac cardioversion were investigated. Reproducible non-PV triggers were mapped using the following self-reference mapping technique. The non-PV trigger reproducibility was defined by 2 criteria: 1) identical intracardiac atrial electrogram sequences and 2) less than a 10 msec fluctuation in the coupling interval between the prior sinus beat and non-PV trigger.

The details of the self-reference mapping have been previously reported [9, 10]. A summary of this technique is as follows. First, the operator was advised to place a catheter from the lateral right atrium, septal right atrium to the SVC, and coronary sinus. After interpreting the trigger activation sequence, the operator placed the PEN catheter at the earliest possible activation site. Using the PEN multipolar catheter, the operator annotated the earliest site of local activation and placed a reference tag there. The multipolar catheter was then moved to the reference tag and the process was repeated. Finally, we identified a cluster of tags. The origin of the non-PV trigger must be in the area of the previous cluster. Local activation around the oldest activation site was evaluated by high density mapping. Radiofrequency ablation was applied with open irrigation contact force catheters. The power setting was 20–35 W. The endpoint of a successful ablation was the non-inducibility of AF.

## Electrophysiological analysis of non-PV triggers

The electrophysiological data at the earliest activation site of AF triggers obtained by the PEN catheter were analyzed. Therefore, the electrogram was obtained by 1-mm electrodes with 2-mm interelectrode spacing. Bipolar electrograms were filtered between 30 and 500 Hz. According to the previous report [12], we evaluated the peak-to-peak electrogram amplitude and defined a low voltage as <0.5mV and normal voltage as ≧0.5mV. The bipolar voltages at the trigger origins were evaluated during sinus rhythm and at the onset of AF. The sinus voltage was assessed at the last sinus rhythm, just before AF initiation. Triggers were divided into 2 groups (normal sinus voltage group and low sinus voltage group) according to the voltage during sinus rhythm. The voltage change ratio was calculated as follows: (voltage change ratio) = (trigger voltage at the onset of AF) / (Sinus voltage of triggers). The trigger coupling interval was evaluated from the onset of the previous sinus beat to the onset of the trigger. An

electrophysiological analysis of non-PV triggers is shown in Fig 1. Non-PV triggers from the SVC or LAPW were excluded from this study.

## Statistical analysis

The continuous variables were presented as the median [interquartile range], and categorical variables as exact numbers and percentages. Continuous variables were compared using a Mann-Whitney U test and Wilcoxon single-rank test. Categorical variables were performed using a Fisher's exact test. The correlation of the continuous variables was assessed using the Pearson correlation coefficient analysis. A P <0.05 was considered statistically significant in all analyzes. The statistical analyses were performed using a commercially available statistical package (JMP Pro14, SAS Institute, Inc., Cary, North Carolina, USA).

## Results

### Clinical characteristics, results of the self-reference mapping, and trigger ablation

A total of 32 non-PV-SVC-LAPW triggers occurred in 23 patients (5%). Polygraph back up data was not available in 1 patient. It was difficult to confirm the earliest trigger site in 1 patient with the synchronization between the CARTO system and polygraph data. The remaining 28 triggers in 21 patients, in whom detailed electrophysiological data were obtained, were enrolled in this study. The patient clinical characteristics are summarized in Table 1. The median age

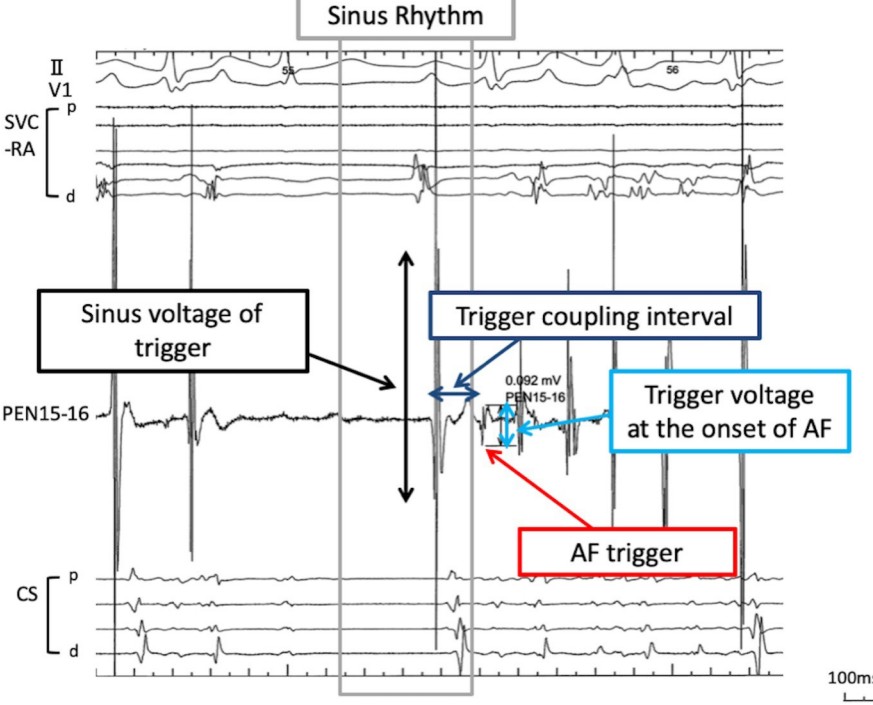

**Fig 1. Illustration of the measurement of the electrophysiological characteristics at the earliest activation site.** The electrogram was obtained by a PentaRay NAV catheter (Biosense Webster Diamond Bar, CA, USA) with 1-mm electrodes with 2-mm interelectrode spacing. The bipolar electrograms were filtered between 30 and 500 Hz. At the earliest activation site of the AF initiating trigger, the peak-to-peak electrogram amplitudes during sinus rhythm and at the onset of AF were measured. The trigger coupling interval was evaluated from the onset of the previous sinus beat to the onset of the trigger. CS, coronary sinus; PEN, PentaRay NAV catheter; RA, right atrium; SVC, superior vena cava.

**Table 1. Patients characteristics and results of self-reference mapping and ablation.**

|  | Values |
|---|---|
| Patients characteristics | 21 patients |
| Age | 69 [61, 75] |
| Male | 15 (71) |
| Paroxysmal/ Persistent (<12 months)/ long-standing (>12 months) | 11/6/4 (52/29/19) |
| Session (1st/ 2nd/ 3rd) | 16/3/2 (76/14/10) |
| Hypertension | 12 (57) |
| Diabetic mellitus | 3 (14) |
| Heart failure | 6 (29) |
| Non-ischemic cardiomyopathy | 2 (10) |
| Ischemic cardiomyopathy | 1 (5) |
| Chronic kidney disease | 4 (19) |
| Left ventricular ejection fraction (%) | 67 [59, 72] |
| Left atrial diameter (mm) | 42 [39, 47] |
| β blocker | 7 (33) |
| ACEI or ARB | 8 (38) |
| Estimated glomerular filtration rate (ml/min/1.73m2) | 66.5 [59.3, 80.9] |
| BNP (pg/ml) | 89.5 [30.2, 152.9] |
| Self-reference mapping results | 21 patients, 28 triggers |
| Number of mapped triggers | |
| N = 1 | 16 (76) |
| N = 2 | 3 (14) |
| N = 3 | 2 (10) |
| Number of mapping points to detect the origin of the trigger | 8 [4.3, 9.8] |
| Number of the cardioversion to detect the origin of the trigger | 8 [5.8, 10] |
| Mapping time to detect the origin of a trigger (min) | 8.9 [2.6, 14.9] |
| Distribution of triggers | |
| Right atrium | 16 (57) |
| Left atrium | 12 (43) |
| Trigger ablation results | 28 triggers |
| Successful trigger elimination | 28 (100) |
| Number of radiofrequency applications for the trigger ablation | 9 [3, 12] |
| Total radiofrequency time for the trigger ablation (min) | 4.3 [2.6, 6.0] |
| Complication | 0 (0) |

Data are given as the n (%) or median [quartile 1, 3]. ACEI = angiotensin converting enzyme inhibitor;
ARB = angiotensin II receptor blocker

was 69, and there were 15 males (71%), 11 with paroxysmal AF (52%), 5 with re-do sessions (24%), 2 with non-ischemic cardiomyopathy (10%), and 1 with ischemic cardiomyopathy (5%). The median left ventricular ejection fraction was 67% and left atrial diameter 42mm.

Multiple non-PV-SVC-LAPW triggers were observed in 5 patients (24%). The median number of self-reference mapping points to detect the trigger origin was 8 [4.3, 9.8]. The median number of the cardioversion to detect the trigger origin was 8 [5.8, 10]. The median mapping time for the self-reference mapping to detect the trigger origin was 8.9 min. The trigger origin was in the right atrium for 16 triggers (57%) and left atrium for 12 (43%). The distribution of non-PV-SVC-LAPW triggers were shown in Fig 2. All non-PV-SVC-LAPW triggers were eliminated by radiofrequency ablation. The median number of radiofrequency

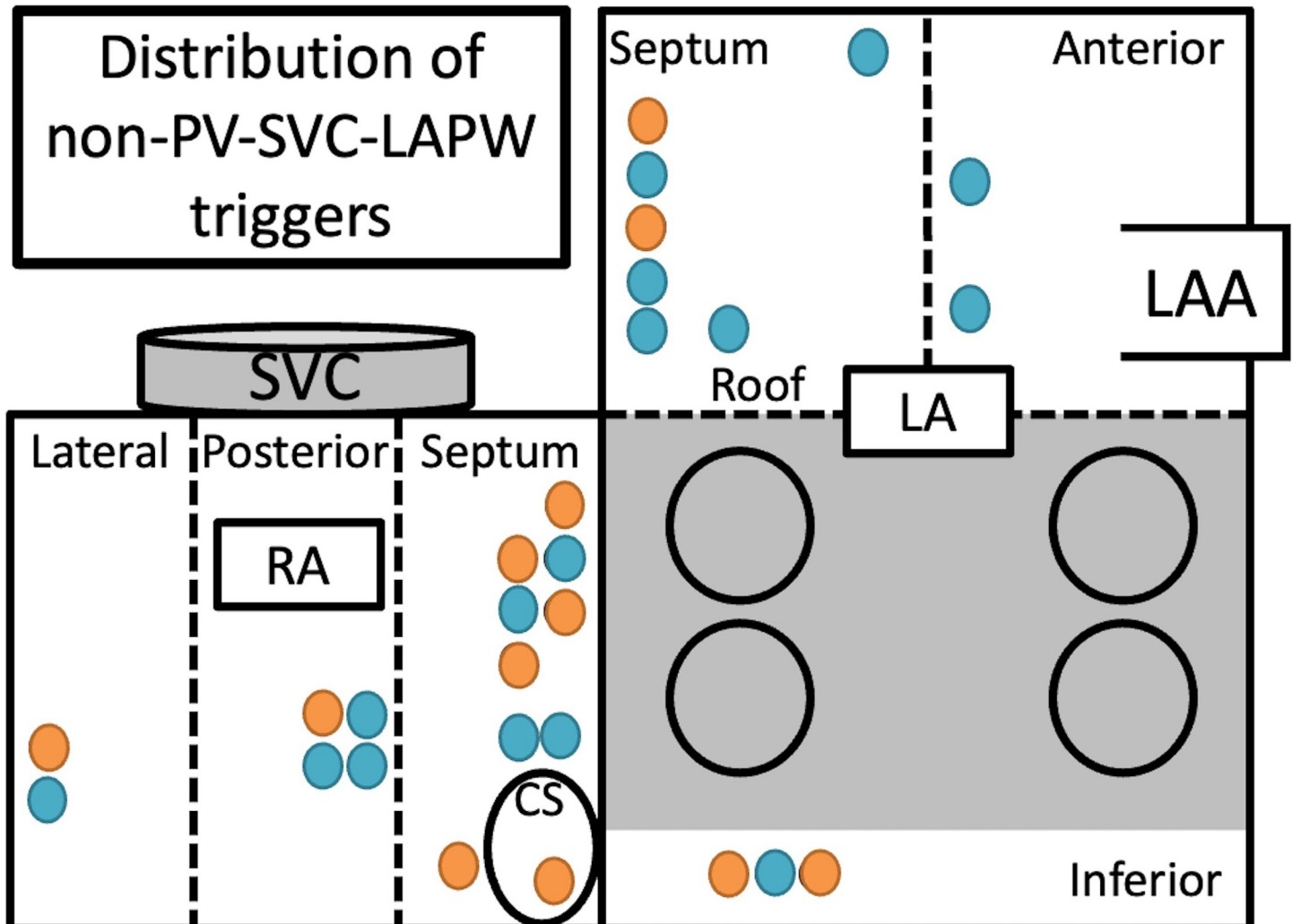

**Fig 2. The distribution of non-PV-SVC-LAPW triggers.** Blue tag indicates normal sinus voltage. Orange tag indicates low sinus voltage. CS = coronary sinus; PV = pulmonary vein; LA = left atrium; RA = right atrium; LAA = left atrial appendage.

applications and time per each trigger were 9 times and 4.3 min, respectively. No major complications related to the AF ablation procedure were observed. The self-reference mapping and trigger ablation results are summarized in Table 1. Among 23 patients, 5 patients had AF/AT recurrences. Three had re-do session, and none of them had recurrences of the non-PV-SVC-LAPW triggers, which was targeted at the index procedure.

## Electrophysiological characteristics of non-PV triggers

The median sinus voltage at the trigger origins was 0.58mV. A low voltage (<0.5mV) was observed at 12 triggers (43%). The median trigger voltage at the onset of AF was 0.16mV. A low voltage at the onset of AF (<0.5mV) was observed in 27 triggers (96%). The median voltage change ratio was 0.35. The median trigger coupling interval between the prior sinus beat and AF initiation was 182msec. A trigger coupling interval of <200 msec was observed at 18 triggers (64%). The electrophysiological characteristics of the non-PV-SVC-LAPW triggers are summarized in Table 2. The trigger voltage at the onset of AF was significantly more reduced

than the sinus voltage (p = 0.005) (Fig 3A). The voltage change ratio was significantly correlated to the trigger coupling interval (R = 0.44, β = 0.44, p = 0.0189) (Fig 3B), whereas the sinus and trigger voltages were not correlated to the trigger interval (p = 0.211 and 0.954, respectively, Fig 3C and 3D).

### Non-centrifugal activation at the onset of AF

An activation with a preferential like conduction turning around was observed at 1 trigger (Fig 4). The earliest potential recorded by electrode pair PEN15-16 did not conduct to PEN13-14, which was on the same spline of the PEN catheter, but instead conducted to PEN 9–10 and then PEN 13–14 was activated. The surrounding atrial muscle, where the other PEN electrodes were placed, was not activated. The earliest potential was a discrete pre-potential and a single point ablation (started with 25W and up to 30W, 22sec) rendered the AF non-inducible.

### Difference in the electrophysiological characteristics between the normal sinus voltage and low sinus voltage groups

The normal sinus voltage group consisted of 16 triggers and low sinus voltage group 12 triggers. The sinus voltage at the trigger origin was higher in the normal sinus voltage group than low sinus voltage group (0.86mv vs. 0.20mV, p<0.001). The distribution of the right or left atrium were comparable between 2 groups (p = 0.459). The trigger voltage at the onset of the AF in the normal sinus voltage group was as low as that in the low sinus voltage group (0.17mV vs. 0.13mV, p = 0.137). The voltage change ratio was significantly lower in the normal sinus voltage group than low sinus voltage group (0.20 vs. 0.60, p = 0.002). The trigger coupling interval between the prior sinus beat and AF initiation was comparable between the 2 groups (170ms vs. 185ms, p = 0.353). The number of mapping points to detect the origin of a trigger (8 times vs. 8 times, p = 0.833) and mapping time (8.9 min vs. 8.2 min, p = 0.944) were comparable between the 2 groups. The number of radiofrequency applications for a trigger (8 times vs. 10 times, p = 0.457) and the total ablation time for a trigger (5.4 min vs. 3.7 min, p = 0.624) were comparable between the 2 groups. These results are summarized in Table 3 and Fig 5.

## Discussion

### Major findings

Our study was the first to describe the detailed electrophysiological characteristics of the non-PV-SVC-LAPW triggers at the successful ablation sites. The major findings in our study were 1) the trigger voltage at the onset of AF was low (median 0.16mV), 2) the coupling interval of the trigger of the non-PV-SVC-LAPW triggers was median 182 msec, and 3) the voltage

**Table 2. Electrophysiological characteristics of non-PV triggers.**

|  | Values |
| --- | --- |
| Sinus voltage at the trigger origin (mV) | 0.58 [0.22, 1.02] |
| Sinus voltage at the trigger origin <0.5mV | 12 (43) |
| Trigger voltage at the onset of AF (mV) | 0.16 [0.11, 0.23] |
| Trigger voltage at the onset of AF <0.5mV | 27 (96) |
| Voltage change ratio | 0.35 [0.15, 0.62] |
| Trigger coupling interval (msec) | 182 [153, 222] |
| Trigger coupling interval <200msec | 18 (64%) |

Data are given as the n (%) or median [quartile1, 3]. AF = atrial fibrillation

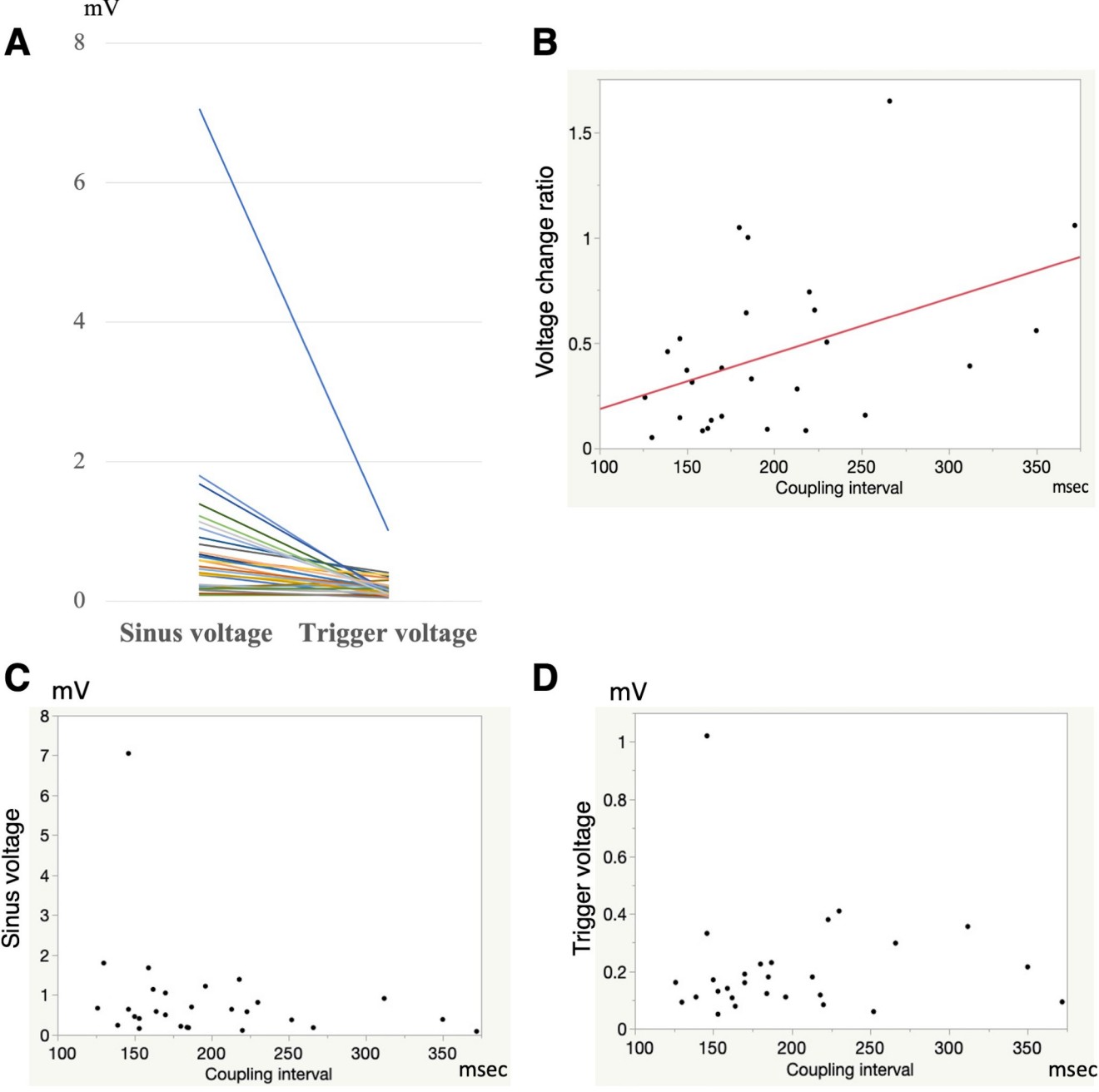

**Fig 3. Left panel:** Voltage change from the sinus beat to the onset of AF. The trigger voltage at the onset of AF was significantly more reduced than the sinus voltage (0.16 mV vs. 0.58mV, p = 0.005). **Right panel:** The relationship between the voltage change ratio and trigger coupling interval. The voltage change ratio was significantly correlated to the trigger coupling interval (R = 0.44, β = 0.44, p = 0.0189).

change ratio was significantly lower in the normal sinus voltage group than low sinus voltage group (0.20 vs. 0.60, p = 0.002).

## Trigger voltage at the onset of the initiation of AF

The trigger voltage at the onset of AF was low (median 0.16mV), even though those electrograms were obtained by a multi-electrode PEN catheter with 1-mm electrodes and a 2-mm

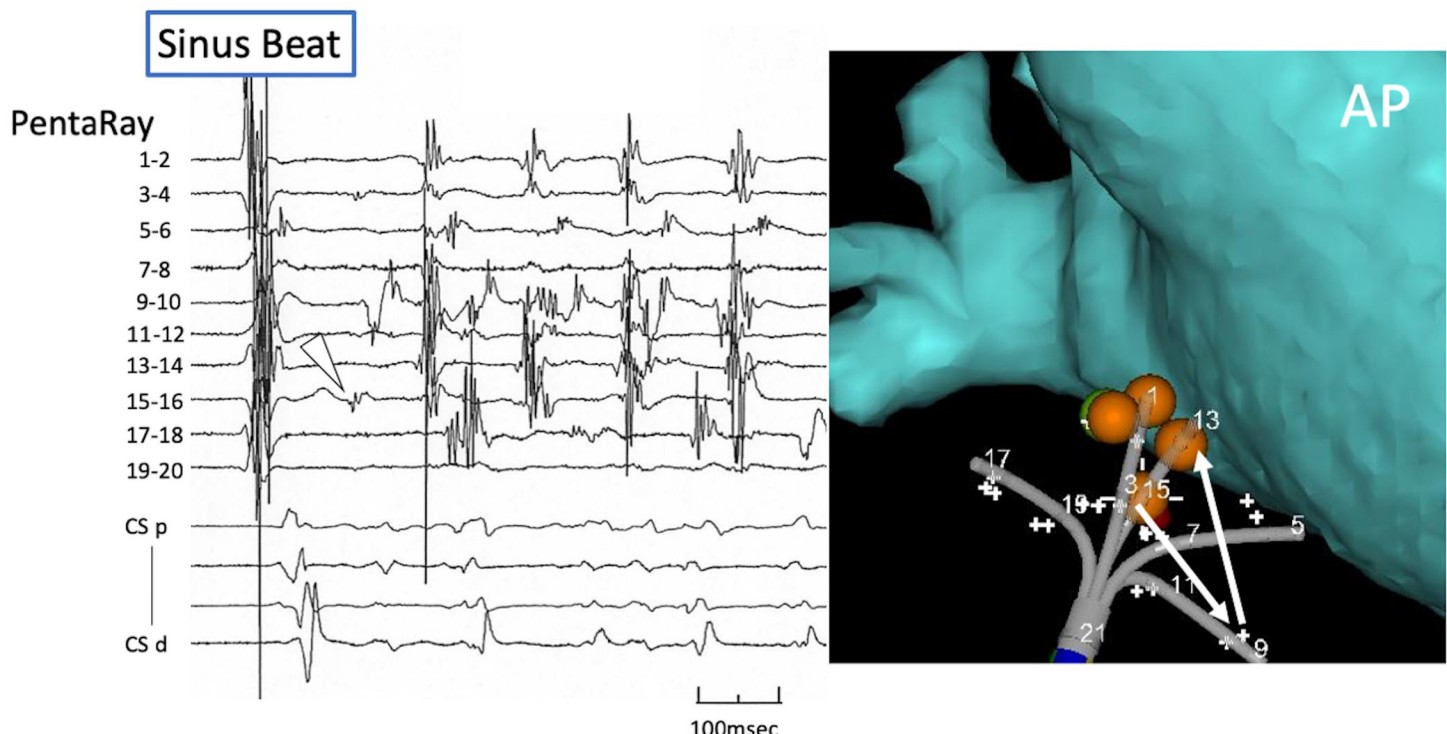

**Fig 4. Left panel:** The intracardiac electrograms during the onset of AF by the non-pulmonary vein trigger. The first beat was sinus rhythm. The earliest potential of the non-pulmonary vein trigger was recorded by PEN15-16 (white arrowhead). The next activated potential was recorded by PEN 9–10 and then PEN 13–14 was activated. **Right panel:** The location of the PEN. The white arrows show the activation sequence of the non-pulmonary vein trigger. AP, anterior-posterior view; CS, coronary sinus; PEN, PentaRay® NAV catheter.

interelectrode spacing. Small and closely spaced electrodes are considered to provide a higher mapping resolution and to identify distinct potentials in low voltage areas in both the atria [13] and ventricles [14] as compared to a larger tip ablation catheter. The self-reference mapping technique uses a PEN multi-electrode catheter to cover a wide area as well as possible to

**Table 3. Differences in the electrophysiological characteristics between a normal voltage and low voltage during sinus rhythm.**

| | Normal Voltage during Sinus Rhythm (n = 16 triggers) | Low Voltage during Sinus Rhythm (n = 12 triggers) | P value |
|---|---|---|---|
| Sinus voltage at the trigger origin (mV) | 0.86 [0.64, 1.35] | 0.20 [0.17, 0.38] | <0.001 |
| Right atrium/ left atrium | 8 / 8 | 8 / 4 | 0.459 |
| Trigger voltage at the onset of the AF (mV) | 0.17 [0.11, 0.35] | 0.13 [0.09, 0.21] | 0.137 |
| Trigger voltage at the onset of the AF <0.5mV | 15 (94) | 12 (100) | 1.000 |
| Voltage change ratio | 0.20 [0.09, 0.39] | 0.60 [0.33, 1.04] | 0.002 |
| Trigger coupling interval (msec) | 170 [149, 216] | 185 [153, 263] | 0.353 |
| Trigger coupling interval <200msec | 11 (69) | 7 (58) | 0.698 |
| Number of mappings to detect the origin of a trigger | 8 [5, 9] | 8 [4, 20] | 0.833 |
| Mapping time to detect the origin of a trigger (min) | 8.9 [2.3, 14.9] | 8.2 [3.4, 14.4] | 0.944 |
| Number of radiofrequency applications for the trigger ablation | 8 [2, 12] | 10 [4, 12] | 0.457 |
| Total radiofrequency time for the trigger ablation (min) | 5.4 [2.7, 7.1] | 3.7 [2.6, 5.2] | 0.624 |

Data are given as the n (%) or median [quartile1, 3]. AF = atrial fibrillation

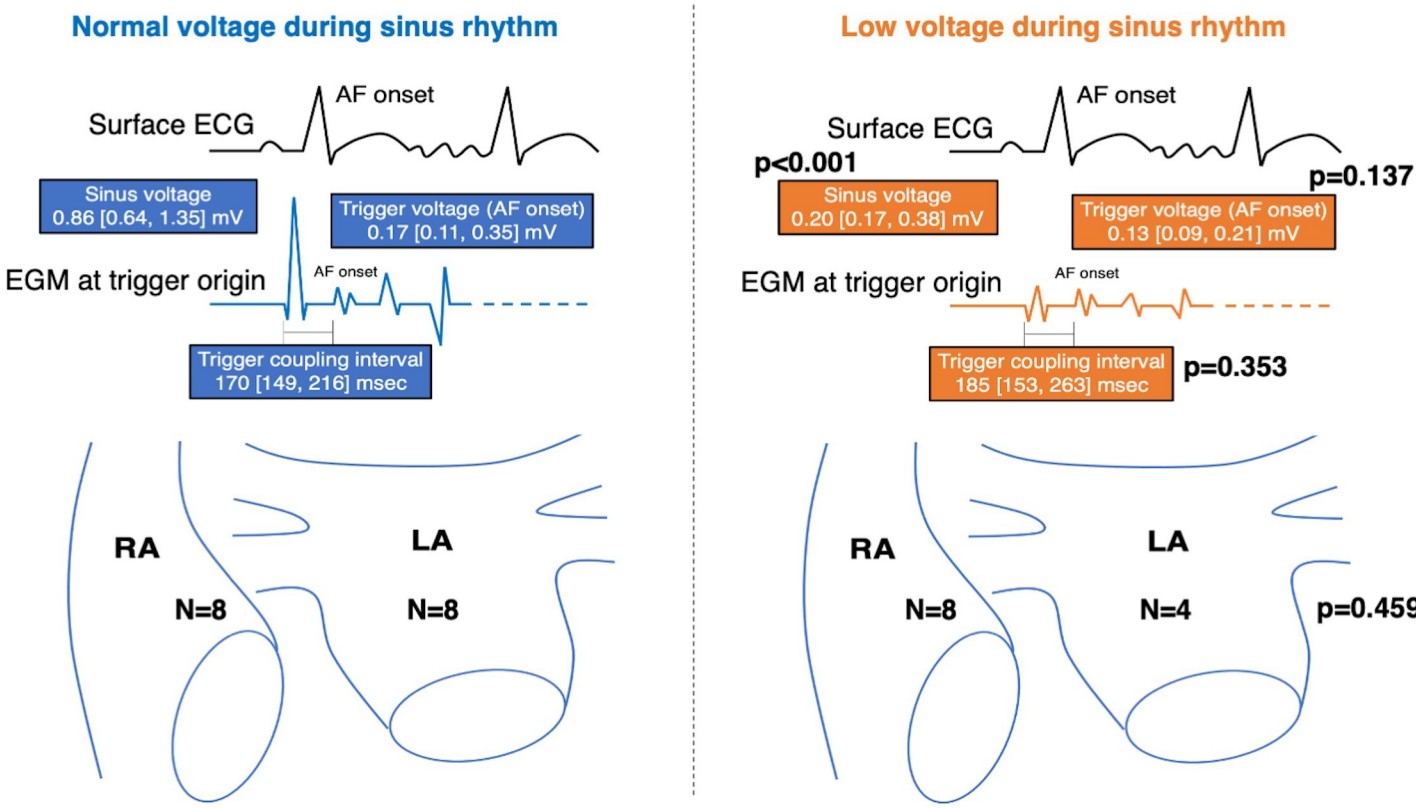

**Fig 5. Difference in the electrophysiological characteristics between the normal sinus voltage and low sinus voltage groups.**

detect any small electrograms at the onset of AF. That concept would relate to good results of a complete elimination of reproducible non-PV-SVC-LAPW triggers. Satisfactory long-term clinical success rates after the non-PV-SVC-LAPW trigger ablation based on the self-reference mapping have already been reported [10] in the same population of this analysis. After the self-reference mapping based non-PV-SVC-LAPW trigger ablation, the AF free survival at 1 year was 82.6%. Three patients who received additional ablation procedures for recurrent AF had late acquired non-PV triggers and the non-PV triggers ablated during the previous procedure have not recurred.

## Non-centrifugal activation at the onset of AF

We previously reported a possible limitation of this self-reference mapping technique in patients after excessive atrial substrate ablation, because excessive ablation would affect the conduction pattern, and the atrial tissue would not conduct centrifugally [9]. However, in this study we could find a non-centrifugal preferential like turning around activation using the self-reference mapping technique. We think the ability to detect low voltage potentials with the small and closely spaced electrodes contributed to this result.

The non-concentric centrifugal activation pattern of PV triggers was previously described using a 64-pole basket catheter [15]. A blocked wavefront introduced a unidirectional conduction and reentrant circuit in the PV. This report would be accordant with our finding of a preferential like turning around activation at the onset of the AF in the non-PV-SVC-LAPW triggers. Moreover, this non-concentric centrifugal activation would result in difficulty in

eliminating non-PV triggers [6, 7] and clinicians should consider the possibility of a preferential like turning around activation with non-PV-SVC-LAPW triggers.

## Coupling interval of the AF triggers

Not all premature atrial complexes induced AF. A short coupling interval trigger can introduce conduction block and contribute to a reentrant circuit. Since reentry is considered to be one of the mechanisms of AF [16], it is reasonable that a short coupling interval trigger can induce AF. Kanda et al. reported that AF inducible ectopy (n = 77) has a short coupling interval (mean coupling interval: 201 msec) and the main origin of the ectopy is the pulmonary veins (n = 74, 96%) [17]. Wang et al. reported that a shorter coupling interval of the P waves in AF initiates premature atrial complexes more than non-AF initiation premature atrial complexes [18]. Their AF initiation triggers were from the PVs and SVC. In our study, the P wave overlapped with the QRS complex or T wave because of a high dose of isoproterenol needed to induce the trigger. Therefore, the P wave interval was not optimal to evaluate in our population. To the best of our knowledge, this is the first study to describe a coupling interval of non-PV-SVC-LAPW triggers.

## Triggers from normal voltages and low voltages during sinus rhythm

There were some possible explanations for the reasons why the voltage change ratio differed between the normal sinus voltage and low sinus voltage groups. First, during the sinus rhythm, the trigger myocardium depolarizes simultaneously with the surrounding myocardium by the propagation from the sinus node. Therefore, the total current at the trigger reflects the simultaneous depolarization of both the trigger myocardium and the surrounding myocardium. However, at the onset of AF, the trigger myocardium depolarizes at first and could only depolarize the neighboring cells according to the source-sink relationship [19], then the total current recorded at the origin becomes low. Second, our evaluation of the voltage was the sum of the surrounding tissues around an electrode. If damaged low voltage tissue was masked by healthy high voltage tissue during the sinus voltage, the voltage obtained by an electrode was high. If AF initiating triggers occurred from damaged tissue as in the low sinus voltage group, a small potential would precede the other potentials. According to the above-described mechanisms, the voltage change ratio was lower in the normal sinus voltage group than low sinus voltage group. Third, the mechanism of the AF initiating triggers differed. Because the target of our study was reproducible triggers with isoproterenol, the possible mechanisms were reentry and triggered activity [16]. Low voltage areas are known to be related to a reentry mechanism in the ventricular tachycardia field [20, 21] and reentry mechanisms are also described for premature ventricular complexes [22]. Triggered activity initiates arrhythmias in the absence of structural heart disease with catecholamine dependency. Depolarization occurs before full repolarization of the fibers, termed early afterdepolarization, and the fibers arise from a reduced level of the membrane potential and exhibit a reduced voltage [16]. Therefore, the voltage change ratio was lower in the normal sinus voltage group with a triggered activity mechanism than in the low sinus voltage group with a reentry mechanism. However, it was difficult to determine which mechanism initiated AF in the normal sinus voltage and low sinus voltage areas, because pacing maneuvers such as entrainment were not possible during AF.

## Clinical implications

Our study demonstrated the detailed electrophysiological characteristics of the non-PV-SVC-LAPW triggers. The triggers at the onset had a low voltage and short coupling interval. The trigger voltage at the onset of AF was low, regardless of whether the sinus voltage of

the trigger was preserved or low. Therefore, electrophysiologists should pay attention not to miss small electrograms. Our results of the trigger coupling interval (median 182msec) would be one of the helpful targets for finding the earliest activation site.

## Limitations

There were limitations to this study. First, we evaluated the electrophysiological characteristics at the earliest activation site as a trigger origin. There was the possibility that we could not find an earlier activated site. However, we could eliminate all of the triggers with a limited number of radiofrequency applications according to the self-reference mapping results. This result would support our mapping quality. Second, the number of non-PV-SVC-LAPW triggers included in this study was small. Limited number of triggers may have affected the result that no triggers from the left atrial appendage or vein of Marshall were observed. However, the low incidence (0.3%) of the left atrial appendage triggers in 7129 patients has been reported [23] and this result was consistent with our data. Third, the electrophysiological characteristics of the non-PV-SVC-LAPW triggers were not compared to other AF triggers, such as triggers from PV, SVC or LAPW. Our strategy to eliminate these triggers were isolation, and the detailed electrophysiological data were not obtained. Forth, we did not create whole voltage map of the right and left atrium. Therefore, assessment of the voltage other than the non-PV-SVC-LAPW trigger origin was possibly not accurate. However, the voltage assessment at the non-PV-SVC-LAPW trigger must be accurate because this mapping provided high density map around the origin and in fact non-PV-SVC-LAPW triggers were all eliminated by the RF application in a small area. Fifth, the unipolar signals were not recorded during the procedure and not assessed in this study. Finally, there is a possibility of AF initiating triggers induced by the mapping catheter. However, to reduce this possibility, the non-PV trigger reproducibility was robustly assessed, comparing the atrial sequences and trigger coupling interval fluctuation, which was permitted only less than 10 msec.

## Conclusion

The trigger voltage at the onset of AF was low, regardless of whether the sinus voltage of the trigger was preserved or low.

## Supporting information

**S1 Table. Dataset of this study.**
(XLSX)

## Acknowledgments

We would like to thank the clinical engineers at Osaka Rosai Hospital and Hoshigaoka Medical Center for their technical support and John Martin's linguistic support.

## Author Contributions

**Conceptualization:** Yasuharu Matsunaga-Lee, Yasuyuki Egami, Masami Nishino.

**Data curation:** Yasuharu Matsunaga-Lee, Sen Matsumoto, Nobutaka Masunaga, Kohei Ukita, Akito Kawamura, Hitoshi Nakamura, Yutaka Matsuhiro, Koji Yasumoto.

**Formal analysis:** Masaki Tsuda.

**Investigation:** Yasuharu Matsunaga-Lee, Sen Matsumoto, Nobutaka Masunaga, Kohei Ukita, Akito Kawamura, Hitoshi Nakamura, Yutaka Matsuhiro, Koji Yasumoto.

**Methodology:** Koji Yasumoto, Naotaka Okamoto.

**Supervision:** Masaki Tsuda, Naotaka Okamoto, Masamichi Yano, Yuzuru Takano, Yasushi Sakata, Masami Nishino, Jun Tanouchi.

**Validation:** Koji Yasumoto.

**Writing – original draft:** Yasuharu Matsunaga-Lee.

**Writing – review & editing:** Yasuyuki Egami, Masamichi Yano, Yuzuru Takano, Yasushi Sakata, Masami Nishino, Jun Tanouchi.

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
