## [Editor Report · Decision Letter 0]

4 Nov 2021

PONE-D-21-34553Electrophysiological Characteristics of Non-Pulmonary Vein Triggers Excluding Origins from the Superior Vena Cava and Left Atrial Posterior Wall: Lessons from the Self-reference Mapping TechniquePLOS ONE

Dear Dr. Nishino,

Thank you for submitting your manuscript to PLOS ONE. Before sending to external reviewers, please clarify several points:

1. How did the authors ensure that PVs were not associated with the AF initiations? Did the authors place PV catheters during determination of the AF origin?

2. How could the authors draw 3D-maps during the initiation of AFs? Did the author also use non-contact mapping to identify the origin (i.e., snap shots)? I think it is difficult to draw 3D-maps during initiation (i.e., only one beat). 

3. Did the authors check voltage maps around the origin of AF? In Fig 1, the signal amplitudes at PEN15-16 keep changing. How were the unipolar signals at the origin of AF?

4. If the authors believe these AFs were not associated with PVs, why did the authors isolate PVs?

We look forward to receiving your revised manuscript.

Kind regards,

Tomohiko Ai, M.D., Ph.D.

Academic Editor

PLOS ONE

Journal Requirements:

Additional Editor Comments:

Before sending to external reviewers, please clarify several points:

1. How did the authors ensure that PVs were not associated with the AF initiations? Did the authors place PV catheters during determination of the AF origin?

2. How could the authors draw 3D-maps during the initiation of AFs? Did the author also use non-contact mapping to identify the origin (i.e., snap shots)? I think it is difficult to draw 3D-maps during initiation (i.e., only one beat).

3. Did the authors check voltage maps around the origin of AF? In Fig 1, the signal amplitudes at PEN15-16 keep changing. How were the unipolar signals at the origin of AF?

4. If the authors believe these AFs were not associated with PVs, why did the authors isolate PVs?
---

## [Author Response · Author response to Decision Letter 0]

28 Nov 2021

Response to Editor/Reviewer

We appreciate your great advice for making our manuscript more sophisticated. We provided our responses to the reviewer’s comments in a blue non-bold font as follows. We hope this revision process has remedied all of your concerns with the original manuscript (OM). The changes and corrections of the OM are shown in a highlighted text in the revised manuscript (RM). We hope our manuscript will be accepted for publication with this version.

Editor Comments

1. How did the authors ensure that PVs were not associated with the AF initiations? Did the authors place PV catheters during determination of the AF origin?

→Thank you for your comment. Before AF induction, we performed PV isolation. Moreover, if the earliest activation site was near the PV, we placed a catheter inside the PV isolation line and checked whether PV isolation remained.

2. How could the authors draw 3D-maps during the initiation of AFs? Did the author also use non-contact mapping to identify the origin (i.e., snap shots)? I think it is difficult to draw 3D-maps during initiation (i.e., only one beat).

→Thank you for your comment. I completely agree with your opinion. We used the self-reference mapping technique, reported in Reference #10. In this technique, we don’t draw an activation map. We place a tag at the earliest activation site among a multi-electrodes catheter, and using the earliest activation site as a reference, we move the multi-electrodes catheter to find earlier activation site as shown in the following figure. This figure was reported in Reference #10 and contained in "Response to Reviewers" file.

3. Did the authors check voltage maps around the origin of AF? In Fig 1, the signal amplitudes at PEN15-16 keep changing. How were the unipolar signals at the origin of AF?

→Thank you for your comment. We did not make voltage maps. However, as shown in “Limitations” section, the voltage assessment at the non-PV-SVC-LAPW trigger must be accurate because this mapping provided high density map around the origin and in fact non-PV-SVC-LAPW triggers were all eliminated by the RF application in a small area. 

As you kindly pointed out, voltage maps were not able to obtain during AF storm, non-regular sinus rhythm, like Fig1. Therefore, we evaluated the sinus voltage just before AF initiation. The unipolar voltage was not recorded during the procedure.

According to your suggestion, we added the following sentences.

Page 8, Line 9-10

“The sinus voltage was assessed at the last sinus rhythm, just before AF initiation.”

Page 18, Line 9-7

“Fifth, the unipolar signals were not recorded during the procedure and not assessed in this study.”

4. If the authors believe these AFs were not associated with PVs, why did the authors isolate PVs?

→Thank you for your comment. According to the guideline (2017 HRS/EHRA/ECAS/APHRS/SOLAECE expert consensus statement on catheter and surgical ablation of atrial fibrillation), PV isolation is the most important first step to treat AF with ablation. Therefore, we performed PV isolation before AF induction in all cases.

---

## [Decision Letter · Decision Letter 1]

5 Jan 2022

PONE-D-21-34553R1Electrophysiological Characteristics of Non-Pulmonary Vein Triggers Excluding Origins from the Superior Vena Cava and Left Atrial Posterior Wall: Lessons from the Self-reference Mapping TechniquePLOS ONE

Dear Dr. Nishino,

Thank you for submitting your manuscript to PLOS ONE. After careful consideration, we feel that it has merit but does not fully meet PLOS ONE’s publication criteria as it currently stands. Therefore, we invite you to submit a revised version of the manuscript that addresses the points raised during the review process.

Your paper was evaluated by three experts in the field. Though the topic is interesting, I think all reviewers are somehow skeptical of the feasibility and accuracy of the mapping method. Further validation is necessary. Please read the comments carefully and address the issues accordingly.      

We look forward to receiving your revised manuscript.

Kind regards,

Tomohiko Ai, M.D., Ph.D.

Academic Editor

PLOS ONE

Reviewers' comments:

Reviewer's Responses to Questions

**Comments to the Author**

1. If the authors have adequately addressed your comments raised in a previous round of review and you feel that this manuscript is now acceptable for publication, you may indicate that here to bypass the “Comments to the Author” section, enter your conflict of interest statement in the “Confidential to Editor” section, and submit your "Accept" recommendation.

Reviewer #1: (No Response)

Reviewer #2: (No Response)

Reviewer #3: (No Response)

2. Is the manuscript technically sound, and do the data support the conclusions?

Reviewer #1: Partly

Reviewer #2: Partly

Reviewer #3: Yes

3. Has the statistical analysis been performed appropriately and rigorously? 

Reviewer #1: Yes

Reviewer #2: I Don't Know

Reviewer #3: I Don't Know

4. Have the authors made all data underlying the findings in their manuscript fully available?

Reviewer #1: Yes

Reviewer #2: Yes

Reviewer #3: Yes

5. Is the manuscript presented in an intelligible fashion and written in standard English?

Reviewer #1: Yes

Reviewer #2: Yes

Reviewer #3: Yes

6. Review Comments to the Author

Reviewer #1: The authors examined the coupling interval and the voltage of the non-PV-SVC-LAPW triggers initiating AF and compared the voltage of them with that of the sinus rhythm. I think this study is interesting. However several issues should be clarified.

1. The authors concluded that Non-PV-SVC-LAPW triggers had a short coupling interval (Page2, Line18). Compared with what was the coupling interval of the AF triggers short? Since it is obvious that the coupling interval of the AF trigger is shorter than the cycle length of the sinus rhythm, the coupling interval of the AF trigger should be compared with those of atrial premature beats not initiating AF that were observed in the 21 patients.

2. The authors described the self-reference mapping technique (Page6, Line4). However, I think that the triggers originating from certain locations such as the CS, region near the tricuspid annulus are cumbersome to map using the PEN catheter alone. Were all the non-PV triggers able to be mapped using the PEN catheter alone without ablation catheter?

3. The authors found that the median number of self-reference mapping points to detect the trigger origin was 8 (Page8, Line12). Was external or internal electrical cardioversion used to restore the sinus rhythm during the mapping? What is the median number of the cardioversion needed to detect the non-PV trigger origin in each patient?

4. The authors showed the activation like preferential conduction (Page9, Line 13). Was the PEN catheter located at the right atrial septum (near the fossa ovalis)? It looks like that the earliest small potential was followed by the large sharp potential at Pen 15-16. The earliest small potential may be caused by ectopy from the LA septum. Was the electrogram of the LA septum simultaneously recorded using the ablation catheter during the mapping?

Reviewer #2: This is a sub-study of their previous report that showed clinical outcomes of their unique technique called self-reference mapping. In the present study, the authors evaluated the electrophysiological characteristics of the AF triggers, excluding origins from the PV, SVC, and LA posterior wall (non-PV-SVC-LAPW).

The authors showed that the non-PV-SVC-LAPW triggers had a short coupling interval and the voltage at the onset of AF was low regardless of the voltage during sinus rhythm.

I found the manuscript to be insightful and well-written. The authors adequately responded to the comments of the previous reviewer. Here, I would like to raise just one but critical issue.

The authors speculated two possible explanations regarding the mechanisms of the voltage change of the trigger site in terms of the voltage during sinus rhythm. They speculated that damaged low voltage tissue might be masked by healthy high voltage tissue during the sinus voltage. I think there were no evidence to support this hypothesis because they did not create whole voltage map and compare the voltage between the trigger and the other intact area.

On the other hand, the authors also presented that the voltage at the onset of AF was consistently low regardless of the voltage during sinus rhythm (Table 3, 0.17 vs. 0.13 mV, p=0.137).

The source-sink relationship could also be the reason for the low voltage at the onset of AF. During the sinus rhythm, the trigger myocardium depolarizes simultaneously with the surrounding myocardium by the propagation from the sinus node. Therefore, the total current at the trigger reflects the simultaneous depolarization of both the trigger myocardium and the surrounding myocardium. However, at the onset of AF, the trigger myocardium depolarizes at first and could only depolarize the neighboring cells, then the total current recorded at the origin becomes low.

I would like to recommend the authors to check the review by Dr. Peter Spector (Circ Arrhythm Electrophysiol. 2013;6:655-661) and consider incorporating this possible mechanism into the discussion.

Reviewer #3: Authors described the electrophysiological characteristics of non-PV trigger excluding the triggers from left atrial posterior wall (LAPW) and superior vena cava (SVC) using the self-reference mapping technique. Thirty-two non-PV-SVC-LAPW triggers were documented in 23 out of 446 patients (5%), and 28 triggers were evaluated in this study. Authors found 1) the trigger voltage at the onset of AF was low, and 2) coupling interval from the non-PV-SVC-LAPW was short. Non-PV-SVC-LAPW trigger appeared from not only the low voltage area but also normal voltage area. Non-PV-SVC-LAPW origin trigger are difficult to map the origin accurately and their electrophysiological characteristics are not well understood. Overall, the manuscript was well written and some findings were very informative to the reader. However, reviewer has some questions and comments.

Major comments

1. Authors divided the non-PV-SVC-LAPW triggers into 2 groups according to the voltage during sinus rhythm. The voltage change ratio was lower in the group with normal voltage during sinus rhythm than the group with low voltage during sinus rhythm. This finding is natural because the voltage of trigger was comparable between 2 groups and authors defined the group according to the voltage during sinus rhythm. What is the difference of electrophysiological significance of the voltage change ratio compared to the voltage during sinus rhythm at the part of the triggers?

2. Authors showed the relationship between the voltage change ratio and trigger coupling interval in Figure 3 right panel. Please create and analyze the diagrams of the relationship between the voltage at the trigger part during sinus rhythm and trigger coupling interval, and the relationship between the voltage of trigger and coupling interval.

3. Please show the outcome of the patients in this study after the procedure. If redo-procedures were performed in recurrent cases, please show the electrophysiological finding of redo-procedure.

Minor comments

1. Authors defined the non-PV trigger as AF initiating triggers and mapped average 8 times to detect the origin of the trigger. That’s means authors required the cardioversion at least average 8 times. Please show the number of cardioversion which authors performed.

2. Please mention the potential of AF initiating triggers induced by the mapping catheter.

7. PLOS authors have the option to publish the peer review history of their article (what does this mean?). If published, this will include your full peer review and any attached files.

Reviewer #1: No

Reviewer #2: No

Reviewer #3: No

---

## [Author Response · Author response to Decision Letter 1]

20 Jan 2022

Response to Reviewer #1

We appreciate your great advice for making our manuscript more sophisticated. We provided our responses to the reviewer’s comments in a blue non-bold font as follows. We hope this revision process has remedied all of your concerns with the original manuscript (OM). The changes and corrections of the OM are shown in a highlighted text in the revised manuscript (RM). We hope our manuscript will be accepted for publication with this version.

Reviewer #1: The authors examined the coupling interval and the voltage of the non-PV-SVC-LAPW triggers initiating AF and compared the voltage of them with that of the sinus rhythm. I think this study is interesting. However, several issues should be clarified.

1. The authors concluded that Non-PV-SVC-LAPW triggers had a short coupling interval (Page2, Line18). Compared with what was the coupling interval of the AF triggers short? Since it is obvious that the coupling interval of the AF trigger is shorter than the cycle length of the sinus rhythm, the coupling interval of the AF trigger should be compared with those of atrial premature beats not initiating AF that were observed in the 21 patients.

→Thank you for your comment. I totally agree with your comment. I should have used “short” comparing to something. As you suggested, we should have compared coupling interval with the non-AF initiating PACs. But we did not map the origin (earliest activation site) of the non-AF initiating PACs. Therefore, it is not possible to compare coupling interval of the AF initiating triggers and the non-AF initiating PACs. We deleted the sentence “Non-PV-SVC-LAPW triggers had a short coupling interval” from the “Conclusions” (Page2, Line18, Page 18, Line 12 in the OM)　and other pages (Page 13, Line 2, Page 15, Line 2, Line 12, Page 17, Line 6 in the OM). 

2. The authors described the self-reference mapping technique (Page6, Line4). However, I think that the triggers originating from certain locations such as the CS, region near the tricuspid annulus are cumbersome to map using the PEN catheter alone. Were all the non-PV triggers able to be mapped using the PEN catheter alone without ablation catheter?

→Thank you for your great comment. It is an important point to do the self-reference mapping. As shown in “Methods/ Non-PV trigger Induction and Self-reference Mapping of non-PV triggers” (Page7, Line11-13), the first step of the mapping is to speculate the earliest activation site by interpreting the atrial sequence of the fixed catheters at the lateral RA, septal RA-SVC and CS. When there were multiple possible earliest activation sites, we compared them using the PEN and ablation catheter or else as shown below figure (Reference #8, Matsunaga-Lee Y, et al. Int J Cardiol 2020; 321: 81-87). After these steps, the last step to find the earliest activation site was performed using the PEN in all cases, including the origins from the CS and near the tricuspid annulus.

3. The authors found that the median number of self-reference mapping points to detect the trigger origin was 8 (Page8, Line12). Was external or internal electrical cardioversion used to restore the sinus rhythm during the mapping? What is the median number of the cardioversion needed to detect the non-PV trigger origin in each patient?

→Thank you for your comment. As you mentioned, we used internal electrical cardioversion to restore sinus during AF trigger mapping. The number of the cardioversion was similar to the number of the mapping, but slightly different because in PAF patients, AF terminated spontaneously. In non-paroxysmal AF patients, they were almost same. According to your comment, we added the following description in “Result” and changed Table1 and Supplemental Table (dataset of this study) to include the information of the cardioversion.

Page 8, line 12-13 in the RM

“The median number of the cardioversion to detect the trigger origin was 8 [5.8, 10].”

Table 1

Self-reference mapping results 21 patients, 28 triggers

Number of mapped triggers 

 N = 1 16 (76)

 N = 2 3 (14)

 N = 3 2 (10)

Number of mapping points to detect the origin of the trigger 8 [4.3, 9.8]

Number of the cardioversion to detect the origin of the trigger 8 [5.8, 10]

Mapping time to detect the origin of a trigger (min) 8.9 [2.6, 14.9]

Distribution of triggers 

 Right atrium 16 (57)

 Left atrium 12 (43)

4. The authors showed the activation like preferential conduction (Page9, Line 13). Was the PEN catheter located at the right atrial septum (near the fossa ovalis)? It looks like that the earliest small potential was followed by the large sharp potential at Pen 15-16. The earliest small potential may be caused by ectopy from the LA septum. Was the electrogram of the LA septum simultaneously recorded using the ablation catheter during the mapping?

→Thank you for your comment. The PEN was located at the RA septum near sinus venosus area, not near the fossa ovalis (FO). However, intracardiac echocardiography was not used to mark the FO. Before locating PEN at this position, we compared the LA and RA septum and found that the RA was earlier activated than the LA. The result that the only single-point RF from the RA side rendered AF non-inducible also supported that the trigger origin was from the RA.

 

Response to Reviewer #2

We appreciate your great advice for making our manuscript more sophisticated. We provided our responses to the reviewer’s comments in a blue non-bold font as follows. We hope this revision process has remedied all of your concerns with the original manuscript (OM). The changes and corrections of the OM are shown in a highlighted text in the revised manuscript (RM). We hope our manuscript will be accepted for publication with this version.

Reviewer #2: This is a sub-study of their previous report that showed clinical outcomes of their unique technique called self-reference mapping. In the present study, the authors evaluated the electrophysiological characteristics of the AF triggers, excluding origins from the PV, SVC, and LA posterior wall (non-PV-SVC-LAPW).

The authors showed that the non-PV-SVC-LAPW triggers had a short coupling interval and the voltage at the onset of AF was low regardless of the voltage during sinus rhythm.

I found the manuscript to be insightful and well-written. The authors adequately responded to the comments of the previous reviewer. Here, I would like to raise just one but critical issue.

The authors speculated two possible explanations regarding the mechanisms of the voltage change of the trigger site in terms of the voltage during sinus rhythm. They speculated that damaged low voltage tissue might be masked by healthy high voltage tissue during the sinus voltage. I think there were no evidence to support this hypothesis because they did not create whole voltage map and compare the voltage between the trigger and the other intact area.

On the other hand, the authors also presented that the voltage at the onset of AF was consistently low regardless of the voltage during sinus rhythm (Table 3, 0.17 vs. 0.13 mV, p=0.137).

The source-sink relationship could also be the reason for the low voltage at the onset of AF. During the sinus rhythm, the trigger myocardium depolarizes simultaneously with the surrounding myocardium by the propagation from the sinus node. Therefore, the total current at the trigger reflects the simultaneous depolarization of both the trigger myocardium and the surrounding myocardium. However, at the onset of AF, the trigger myocardium depolarizes at first and could only depolarize the neighboring cells, then the total current recorded at the origin becomes low.

I would like to recommend the authors to check the review by Dr. Peter Spector (Circ Arrhythm Electrophysiol. 2013;6:655-661) and consider incorporating this possible mechanism into the discussion.

→Thank you for your great recommendation. I agree with your hypothesis. The source-sink mismatch could be the best reason for the low voltage at the AF initiation. I adopted this hypothesis as a first possible mechanism in the “Discussion” as follows.

Page 13, line 17〜page 14, line 3 in the RM

“During the sinus rhythm, the trigger myocardium depolarizes simultaneously with the surrounding myocardium by the propagation from the sinus node. Therefore, the total current at the trigger reflects the simultaneous depolarization of both the trigger myocardium and the surrounding myocardium. However, at the onset of AF, the trigger myocardium depolarizes at first and could only depolarize the neighboring cells according to the source-sink relationship [19], then the total current recorded at the origin becomes low.”

New reference #19

Spector P. Principles of cardiac electric propagation and their implications for re-entrant arrhythmias. Circ Arrhythm Electrophysiol. 2013;6(3):655-61. Epub 2013/06/20. doi: 10.1161/CIRCEP.113.000311. PubMed PMID: 23778249.

 

Response to Reviewer #3

We appreciate your great advice for making our manuscript more sophisticated. We provided our responses to the reviewer’s comments in a blue non-bold font as follows. We hope this revision process has remedied all of your concerns with the original manuscript (OM). The changes and corrections of the OM are shown in a highlighted text in the revised manuscript (RM). We hope our manuscript will be accepted for publication with this version.

Reviewer #3: Authors described the electrophysiological characteristics of non-PV trigger excluding the triggers from left atrial posterior wall (LAPW) and superior vena cava (SVC) using the self-reference mapping technique. Thirty-two non-PV-SVC-LAPW triggers were documented in 23 out of 446 patients (5%), and 28 triggers were evaluated in this study. Authors found 1) the trigger voltage at the onset of AF was low, and 2) coupling interval from the non-PV-SVC-LAPW was short. Non-PV-SVC-LAPW trigger appeared from not only the low voltage area but also normal voltage area. Non-PV-SVC-LAPW origin trigger are difficult to map the origin accurately and their electrophysiological characteristics are not well understood. Overall, the manuscript was well written and some findings were very informative to the reader. However, reviewer has some questions and comments.

Major comments

1. Authors divided the non-PV-SVC-LAPW triggers into 2 groups according to the voltage during sinus rhythm. The voltage change ratio was lower in the group with normal voltage during sinus rhythm than the group with low voltage during sinus rhythm. This finding is natural because the voltage of trigger was comparable between 2 groups and authors defined the group according to the voltage during sinus rhythm. What is the difference of electrophysiological significance of the voltage change ratio compared to the voltage during sinus rhythm at the part of the triggers?

2. Authors showed the relationship between the voltage change ratio and trigger coupling interval in Figure 3 right panel. Please create and analyze the diagrams of the relationship between the voltage at the trigger part during sinus rhythm and trigger coupling interval, and the relationship between the voltage of trigger and coupling interval.

→Thank you for your great comment #1 and #2. I think it is preferable to reply to comments at the same time. I added the relationship between the sinus/trigger voltage and trigger coupling interval as follows. Also, we added Fig 3 C and D. To answer your comment #1, this analysis was important. The sinus/trigger voltage were not correlated to the trigger coupling interval, however voltage change ratio was correlated to the coupling interval. Therefore, even there was an adequate amount of myocardium during long coupling interval (sinus rhythm), only small amount of myocardium could activated at the small coupling interval (AF initiation). This might be explained by the mechanism of the triggered activity (Reference #16, Zipes DP. Mechanisms of clinical arrhythmias. J Cardiovasc Electrophysiol. 2003;14(8):902-12.), as shown in the “Trigger from normal voltages and low voltages during sinus rhythm” in the Discussion part. This is why we analyzed the voltage change ratio, not only the sinus/trigger voltage.

Page 9, line 12-14 in the RM

“The voltage change ratio was significantly correlated to the trigger coupling interval (R=0.44, β=0.44, p=0.0189) (Fig 3 B), whereas the sinus and trigger voltages were not correlated to the trigger interval (p=0.211 and 0.954, respectively, Fig 3 C and D).”

3. Please show the outcome of the patients in this study after the procedure. If redo-procedures were performed in recurrent cases, please show the electrophysiological finding of redo-procedure.

→Thank you for your suggestion. I also wanted to add the outcomes including the re-do procedure. However, in the initial report of the self-reference mapping (Reference #10, Matsunaga-Lee Y, et al. Int J Cardiol 2020; 321: 81-87), we have already reported the clinical outcomes and re-do procedure. (Shortly summarized, 5 patients had AF/AT recurrences. Three had re-do session, and none of them had recurrences of the non-PV-SVC-LAPW triggers, which was targeted at the index procedure.) If we included these data, there might be large overlapping description between current and initial report. Therefore, we were afraid to write them again and added the shortly summarized description in the “Result”.

Page 8, line 19〜page 9, line 2 in the RM

“Among 23 patients, 5 patients had AF/AT recurrences. Three had re-do session, and none of them had recurrences of the non-PV-SVC-LAPW triggers, which was targeted at the index procedure.” 

Minor comments

1. Authors defined the non-PV trigger as AF initiating triggers and mapped average 8 times to detect the origin of the trigger. That’s means authors required the cardioversion at least average 8 times. Please show the number of cardioversion which authors performed.

→Thank you for your advice. As you mentioned, we used internal electrical cardioversion to restore sinus during AF trigger mapping. The number of the cardioversion was similar to the number of the mapping, but slightly different because in PAF patients, AF terminated spontaneously. In non-paroxysmal AF patients, they were almost same. According to your recommendation, we added the following description in “Result” and changed Table1 and Supplemental Table (dataset of this study) to include the information of the cardioversion.

Page 8, line 12-13 in the RM

“The median number of the cardioversion to detect the trigger origin was 8 [5.8, 10].”

Table 1

Self-reference mapping results 21 patients, 28 triggers

Number of mapped triggers 

 N = 1 16 (76)

 N = 2 3 (14)

 N = 3 2 (10)

Number of mapping points to detect the origin of the trigger 8 [4.3, 9.8]

Number of the cardioversion to detect the origin of the trigger 8 [5.8, 10]

Mapping time to detect the origin of a trigger (min) 8.9 [2.6, 14.9]

Distribution of triggers 

 Right atrium 16 (57)

 Left atrium 12 (43)

2. Please mention the potential of AF initiating triggers induced by the mapping catheter.

→Thank you for your comment. I added following sentences in the “Limitations”.

Page 16, line 8-12 in the RM

“Finally, there is a possibility of AF initiating triggers induced by the mapping catheter. However, to reduce this possibility, the non-PV trigger reproducibility was robustly assessed, comparing the atrial sequences and trigger coupling interval fluctuation, which was permitted only less than 10 msec.”

---

## [Decision Letter · Decision Letter 2]

31 Jan 2022

Electrophysiological Characteristics of Non-Pulmonary Vein Triggers Excluding Origins from the Superior Vena Cava and Left Atrial Posterior Wall: Lessons from the Self-reference Mapping Technique

PONE-D-21-34553R2

Dear Dr. Nishino,

We’re pleased to inform you that your manuscript has been judged scientifically suitable for publication and will be formally accepted for publication once it meets all outstanding technical requirements.

Kind regards,

Tomohiko Ai, M.D., Ph.D.

Academic Editor

PLOS ONE

Additional Editor Comments (optional):

Reviewers' comments:

Reviewer's Responses to Questions

**Comments to the Author**

1. If the authors have adequately addressed your comments raised in a previous round of review and you feel that this manuscript is now acceptable for publication, you may indicate that here to bypass the “Comments to the Author” section, enter your conflict of interest statement in the “Confidential to Editor” section, and submit your "Accept" recommendation.

Reviewer #1: All comments have been addressed

Reviewer #2: All comments have been addressed

Reviewer #3: All comments have been addressed

2. Is the manuscript technically sound, and do the data support the conclusions?

Reviewer #1: Yes

Reviewer #2: Yes

Reviewer #3: Yes

3. Has the statistical analysis been performed appropriately and rigorously? 

Reviewer #1: Yes

Reviewer #2: I Don't Know

Reviewer #3: Yes

4. Have the authors made all data underlying the findings in their manuscript fully available?

Reviewer #1: Yes

Reviewer #2: Yes

Reviewer #3: Yes

5. Is the manuscript presented in an intelligible fashion and written in standard English?

Reviewer #1: Yes

Reviewer #2: Yes

Reviewer #3: Yes

6. Review Comments to the Author

Reviewer #1: Thank you for the revision. I think the conclusions has been revised to be reasonable. I have no further comment.

Reviewer #2: The authors have adequately responded to the comment and adopted the source-sink mismatch as the first possible mechanism of the voltage change of the trigger site.

Reviewer #3: I appreciate the authors significant efforts to clarify and address this reviewer's questions. Hopefully the revised manuscript has been strengthened as a result. I have no further comments.

7. PLOS authors have the option to publish the peer review history of their article (what does this mean?). If published, this will include your full peer review and any attached files.

Reviewer #1: No

Reviewer #2: **Yes: **Taku Nishida

Reviewer #3: No

---

## [Editor Report · Acceptance letter]

28 Mar 2022

PONE-D-21-34553R2 

Electrophysiological Characteristics of Non-Pulmonary Vein Triggers Excluding Origins from the Superior Vena Cava and Left Atrial Posterior Wall: Lessons from the Self-reference Mapping Technique 

Dear Dr. Nishino:

I'm pleased to inform you that your manuscript has been deemed suitable for publication in PLOS ONE. Congratulations! Your manuscript is now with our production department. 

Kind regards, 

on behalf of

Dr. Tomohiko Ai 

Academic Editor

PLOS ONE